# The Effect of Hypotensive Anesthesia on Hemoglobin Levels during Total Knee Arthroplasty

**DOI:** 10.3390/jcm10010057

**Published:** 2020-12-26

**Authors:** Tom Schmidt-Braekling, Enrique Goytizolo, Wenzel Waldstein, Nigel Sharrock, Friedrich Boettner

**Affiliations:** 1Adult Reconstruction & Joint Replacement Division, Hospital for Special Surgery, Weill Medical College of Cornell University, New York, NY 10021, USA; tom.schmidt-braekling@ukmuenster.de; 2Department of Orthopedics and Tumor Orthopedics, University of Muenster, 48149 Muenster, Germany; 3Department of Anesthesiology, Hospital for Special Surgery, New York, NY 10021, USA; GoytizoloE@HSS.EDU (E.G.); sharrockn@hss.edu (N.S.); 4Department of Orthopaedics, Vienna General Hospital, Medical University of Vienna, 1090 Vienna, Austria; wwaldstein@gmail.com

**Keywords:** hypotensive anesthesia, total knee arthroplasty, hemoglobin levels, blood transfusions, blood-saving modality

## Abstract

Introduction: Hypotensive epidural anesthesia (HEA) is used in total joint arthroplasty as a safe and effective blood-saving modality. In order to maintain the blood pressure and heart rate patients, receive 1000 to 1500 mL of lactated Ringer’s solution during surgery. While HEA reduces the intraoperative blood loss, the effect of intravenous fluid loading on hemoglobin levels is not fully understood. The current study investigates the effect of HEA on perioperative hemoglobin levels. Materials and Methods: The study included 35 patients operated on by a single surgeon undergoing primary total knee arthroplasty under HEA. Intraoperatively, at least 300 mL of intravenous fluid were given every 15 min over the first 60 min after HEA. Blood samples were drawn before entering the operating room, after HEA, as well as after inflation of the tourniquet, every 15 min thereafter, as well as in the recovery room and on postoperative days one and two. In addition, fluid in- and outtake was recorded. Results: Patients received a mean 1275 mL during the 60 min of tourniquet time. The mean arterial pressure (MAP) 5 min after HEA dropped to 60 mmHg and reached a constant level of around 58 mmHg 15 min after HEA. The average hemoglobin level dropped from 13.9 g/dL prior to HEA, to 12.5 g/dL immediately after HEA (*p* < 0.001). Intraoperatively the hemoglobin level dropped further and reached 11.8 g/dL at 60 min in the absence of blood loss. Conclusions: Hypotensive epidural anesthesia and the resulting fluid substitution resulted in an average hemoglobin drop of 2.1 g/dL within the first 60 min. This needs to be taken into account when evaluating the need for blood transfusions after primary joint replacement surgery under HEA.

## 1. Introduction

Total knee arthroplasty (TKA) is a highly successful treatment option for end-stage arthritis of the knee, however it can result in significant blood loss [1,2]. The average blood loss is up to 1500 mL and the average drop in hemoglobin level is reported to be 3.85 g/dL [1,3]. However, anemia and intraoperative blood loss in particular are considered predictors of postoperative outcome [4,5]. In this context, postoperative anemia is not only associated with a poorer outcome, but also with an increase in the complication rate and a prolonged hospital stay [4,5]. Nevertheless, if it is necessary to transfuse a red cell concentrate; due to symptomatic postoperative anemia, there are risks for the patient. Studies have reported that postoperative blood transfusions after elective arthroplasty of the knee and hip not only increase the risk of serious complications and surgical infections, but might also lead to an increase in mortality [4,5]. 

Hypotensive epidural anesthesia (HEA) is a readily available and effective method to reduce intraoperative blood loss [6]. The technique is performed at an upper thoracic level such as T4, effectively blocking the thoracic sympathetic nerve system. This results in vasodilation and hypotension without reflex tachycardia as the cardiac sympathetic innervation is also blocked. Continuous intravenous infusion of a low-dose epinephrine solution is used to stabilize heart rate, central-venous-pressure (CVP) and cardiac output during hypotension [7]. Lowering the mean arterial pressure (MAP) to 50 mmHg has been found to decrease intraoperative blood loss and transfusion requirements [8,9]. The technique does not appear to adversely affect cardiac, renal or cerebral function and is used safely in patients with hypertension, ischemic heart disease and in the elderly [10,11,12,13,14]. 

During HEA, patients undergoing primary total hip replacement require an average of 1000–1500 mL of lactated Ringer’s solution to control hypotension [7]. Fluid substitution in the setting of even moderate epidural-induced hypotension has been associated with increased blood dilution [15,16]. While the blood volume increased by 25 to 35% of the amount of intravenous fluid given to normotensive patients, double the amount is retained when the patient is hypotensive [17]. The drop in hemoglobin secondary to fluid substitution during HEA needs to be taken into account when determining postoperative transfusion requirements.

The current study investigates the effect of intraoperative fluid substitution during HEA on hemoglobin levels, during and after primary TKA.

## 2. Materials and Methods

Between January and August 2014, thirty-five patients who underwent primary total knee arthroplasty were included in this prospective study. Out of these patients, 3 patients had to be excluded because of inadequate hypotensive anesthesia, resulting in too high blood pressure levels during surgery, leaving 32 patients, 22 female and 10 male patients for inclusion in the study. The senior author performed all surgical procedures. The institutional review board approved the study and all patients gave written consent prior to enrolling on the study. 

Demographic data (including gender, patient age and body mass index), side of surgery, American Society of Anesthesiologists (ASA) score, diagnosis, procedure time, intraoperative blood loss in mL, blood pressure during surgery, administered intravenous fluid in mL every 15 min during the first 60 min of the procedure, hemoglobin and hematocrit levels and the amount of urine output and drain output were recorded.

The first blood sample was drawn from an arterial line in the holding area prior to anesthesia. The patient was then transferred to the operating room and hypotensive epidural anesthesia (HEA) was performed aiming for an upper thoracic level such as T4 sensory block. 5 mL of blood were drawn from the arterial line to measure hemoglobin and hematocrit levels after HEA prior to inflation of the tourniquet and 15 min, 30 min, 45 min and 60 min after tourniquet inflation. In addition, blood was drawn immediately upon arrival in the postoperative care unit as well as postoperative day (POD) 1. Patients were required to receive at least 300 mL of lactate ringer every 15 min of tourniquet time.

Blood pressure and heart rate were recorded preoperatively, prior to HEA, and every 5 min until the end of the procedure. In addition, the intravenous fluids were documented at 15 min intervals during the first 60 min, and every hour thereafter until POD 3. Furthermore the intraoperative blood loss and urine and drain output were recorded until discharge.

Patients with non-inflammatory degenerative joint disease, between the ages of 40 and 80 years and undergoing primary total knee arthroplasty were included. Patients underwent neuraxial anesthesia and had to be rated grade 1 to 3, as per the American Society of Anesthesiologists (ASA) Physical Classification System. Exclusion criteria were patient refusing the participation in the study, contraindication to combined spinal epidural anesthesia (history of lumbar spinal fusion, bleeding disorder, use of clinically relevant anticoagulant or antiplatelet medications, anatomic abnormalities, infection at a potential injection site), patients with congestive heart failure (at least one medication to treat congestive heart failure), coronary artery disease (s/p bypass, stent or AMI), kidney insufficiency (creatinine > 1.5 mg/dL), aortic or mitral valve disease and pulmonary hypertension. In addition, patients were excluded if they received less than 1200 mL of intravenous fluid during 60 min tourniquet time and if HEA resulted in inadequate levels of hypotension.

The tourniquet was used during the first 60 min of the procedure. Therefore, there was no blood loss during the first 60 min of the surgery. In conclusion, the drop in hemoglobin level should be primarily related to the decrease in blood pressure and hemodilution secondary to fluid loading during HEA. The tourniquet always remained in place for 60 min.

### Statistical Analysis

The distributions of variables were tested in exploratory data analysis. The Kolmogorov–Smirnov test was performed to test for the normal distribution of variables. Absolute mean values and differences of hemoglobin were expressed in grams per deciliter with standard deviation (SD). Hematocrit levels were described in volume percentage (vol.%) with SD. As not all variables met the criteria for normal distribution, the Wilcoxon signed ranks were used to compare differences of variables in time. Spearman rank correlation (r_s_) was used for nonparametric correlations. *p*-values < 0.05 were considered significant. Statistical tests were carried out using SPSS 22 software for windows (SPSS Inc., Chicago, Illinois).

## 3. Results

The average age at the time of surgery was 66 years (range: 44 to 79 years) and the average BMI was 30.6 kg/m^2^ (range 20.7 to 42.4 kg/m^2^). Sixty-nine percent of the patients were female (*n* = 22), thirty-one percent male (*n* = 10). Twelve patients (37%) underwent left and 20 patients (63%) right TKA. In all patients, the ASA (American Society of Anesthesiologists) score was assessed, two patients were rated ASA 1, 22 patients ASA 2 and 8 patients ASA 3. All three patients, whom had to be excluded from the study because of insufficient drop of the blood pressure, belonged to ASA 3 group. Mean hemoglobin and hematocrit levels are displayed in Table 1.

Patients received a mean of 1275 mL (SD 78, range: 1150–1500) lactated Ringer’s solution during surgery, of which 440 mL were given preoperatively. Altogether, a mean of 2947 mL (SD 677, range: 1700–4275) lactated Ringer’s solution were substituted at the day of surgery (Table 2).

The mean arterial pressure (MAP) prior to HEA was 87 mmHg. Five minutes after HEA, the mean MAP had dropped to an average of 60 mmHg and 15 min after HEA, the mean MAP dropped to an average of 58 mmHg (Table 2). 

After initiation of hypotensive epidural anesthesia, hemoglobin levels dropped by a mean of 1.4 g/dL (SD 0.5) and hematocrit levels dropped by a mean of 4.3% (SD 2.2). The differences of hemoglobin and hematocrit levels prior and after initiation of HEA were highly significant (*p* < 0.001). 

The mean hemoglobin levels continued to drop during the 60 min of tourniquet time by an average of 0.7 g/dL (SD 0.4), the mean hematocrit dropped by an addition 2.1% (SD 1.3), respectively (*p* < 0.001). The release of the tourniquet resulted in a slight increase of Hb (0.2 g/dL, SD 0.68; *p* = 0.042) and hematocrit (0.6%, SD 1.4; *p* = 0.045) compared to levels 60 min after tourniquet inflation. In addition, the current study reports no difference at any timepoint regarding the intraoperative drop in hemoglobin and hematocrit in patients with a BMI < 30 and >30 kg/m^2^.

The average drain output until POD 1 was 334 mL (range: 0–1100 mL). The mean difference in hemoglobin levels comparing POD 1 and POD 2 was 0.7 g/dL (SD 0.9 g/dL), the mean Hematocrit difference was 2.5% (SD 2.5%), respectively. The mean total drain volume was 334 mL (range: 0–1100). The mean difference in hemoglobin levels between time in PACU and POD 2 significantly correlated with the total drain volume (r_s_ = 0.575, *p* = 0.001). The mean difference of hematocrit levels between PACU and POD 2 also correlated well with the total drain volume (r_s_ = 0.518, *p* = 0.005). Excellent correlations of preoperative hemoglobin levels and hemoglobin levels prior to tourniquet inflation with Hb levels on postoperative day 1 and postoperative day 2 were observed (Table 3).

We compared hematocrit at the different intraoperative timepoints for the respective subgroups (male vs. female, BMI > 30 kg/m^2^, age under 60/67 years) using student’s t-test for independent samples with parametric data, no significant differences were shown. Furthermore, the length of surgery showed very little variance with a mean of 61 (range 60–70) and the 95th percentile was 67 min with no significant differences.

## 4. Discussion

Hypotensive epidural anesthesia and its fluid substitution resulted in an average hemoglobin drop of 2.1 g/dl within the first 60 min after HEA. This needs to be taken into account when evaluating the need for blood transfusions after primary joint replacement surgery under HEA.

The current study has the following limitations: (1) since the blood volume is not measured it is not clear whether hemodilution is the underlying mechanism for the hemoglobin drop, (2) hemoglobin levels are determined from an arterial line and this could result in dilution of the blood samples. However, all samples were retrieved using the same technique, (3) the application of the tourniquet could result in changes of the overall blood volume, however, exsanguination of the leg should not have an impact on hemoglobin or hematocrit levels. The slight increase in hemoglobin and hematocrit after the release of the tourniquet suggests mixture of non-diluted blood from the operated extremity with diluted blood in the remaining body. (4) After the release of the tourniquet changes in blood loss are likely the result of bleeding from the surgical site and decreased hemodilution after discontinuation of the HEA.

In order to maintain blood pressure and heart rate, the patients received a mean of 1275 mL lactated ringer solution during surgery. This acute volume loading had an effect on the hemoglobin level during the use of the tourniquet. The immediate hemoglobin drop after initiation of the HEA in the absence of blood loss was statistically significant (1.4 g/dL; *p* < 0.001), however, continued changes in hemoglobin levels despite continued fluid substitution remained rather small (0.7 g/dL). The data suggest that initially after the HEA a considerable amount of fluid is retained or mobilized from the extravascular space to balance out the hypotensive effect of a high epidural anesthesia and its blockage of the sympathetic nerve system. The continued infusion of epinephrine probably minimizes the hemodilution associated with continued intravenous fluid infusion and might be the main reason for the reduced effect of HEA on hemoglobin levels later in surgery. After the release of the tourniquet, the hemoglobin level slightly increased by 0.2 g/dL, which might be explained by the mixture of undiluted blood from the operated extremity following the release of the tourniquet. 

HEA has a number of clinical benefits for patients undergoing primary total joint arthroplasty. Lowering the mean arterial pressure (MAP) to 50 mmHg has been found to decrease intraoperative blood loss by up to 40% [9] and reduce postoperative drainage blood volume [8]. HEA can be used in patients with hypertension, ischemic heart disease and in the elderly due to consistent central venous pressure, stroke volume and cardiac output. In addition, the cardiac, renal or cerebral functions are not affected by HEA [6]. 

Sharrock et al. suggest that HEA does not increase the risk for acute renal dysfunction [18]. HEA has been associated not only with reduced intraoperative blood loss, but also with fewer perioperative blood transfusions [6,13,14], a lower rate of deep vein thrombosis [10,11], and a low perioperative mortality rate have been reported [12]. In addition, by minimizing intraoperative blood loss, the surgical exposure is improved.

A randomized study comparing HEA and spinal anesthesia in patients scheduled for TKA reported no differences in renal function and no cardiac or neurologic complications. In addition, no increase in complications has been reported for the elderly [19].

The most recent study concerning HEA in orthopedic surgery reported less blood loss, fewer transfusions and no increase in complications in sacral and pelvic bone tumor resections [20].

Reports on negative side effects of HEA are rare, and include temporary decline in postoperative cognitive, hepatic or renal function [21]. In addition, Liguori et al. reported seven cases of severe bradycardia and five cases of asystole during orthopedic surgery under HEA [22].

While the perioperative anemia secondary to HEA has to be taken into consideration when judging the need for blood transfusion, its overall extend does not seem to outweigh its reported benefits on blood loss and morbidity. 

## 5. Conclusions

While hypotensive epidural anesthesia (HEA) has a significant immediate effect on patients’ hemoglobin levels; the continued drop in hemoglobin, although statistically significant, is rather low. Approximately 2.1 g/dL drop in hemoglobin in the first 60 min after surgery appear to be the result of the hemodilution from the intravenous fluid substitution during HEA. This needs to be taken into consideration when assessing the need for perioperative blood transfusions in the first 24 h after surgery performed under HEA.

## Figures and Tables

**Table 1 jcm-10-00057-t001:** Changes in hemoglobin (g/dL) and hematocrit after initiation of a hypotensive spinal-epidural anesthesia (HSEA). ^a^ Post Anesthesia Care Unit (PACU).

	Prior to HSEA (*n* = 32)	After HSEA (*n* = 32)	15min (*n =* 32)	30min (*n =* 32)	45min (*n =* 32)	60min (*n =* 32)	PACU ^a^ (*n =* 30)	POD1 (*n =* 31)	POD2 (*n =* 31)
**Hemoglobin**	**13.9** (SD 1.3; range: 11.9–16.6)	**12.5** (SD 1.2; range: 10.4–14.9)	**12.2** (SD 1.2; range: 10.2–14.8)	**12.0** (SD 1.2; range: 9.8–14.3)	**11.9** (SD 1.595; range: 9.5–14.6)	**11.8** (SD 1.3; range: 9.4–14.6)	**12.0** (SD 1.0; range: 10.0–14.0)	**11.6** (SD 1.2; range: 9.2–14.6)	**11.0** (SD 1.3; range: 8.9–14.1)
**Hematocrit**	**41.1** (SD 3.46; range: 35.5–47.7)	**36.8** (SD 3.2; range: 30.9–45.0)	**36.0** (SD 3.1; range: 30.6–44.0)	**35.4** (SD 2.9; range: 29.8–42.1)	**35.1** (SD 3.1; range: 28.9–42.8)	**34.7** (SD 3.1; range: 28.7–42.5)	**35.1** (SD 2.7; range: 29.3–41.1)	**35.3** (SD 3.4; range: 29.2–43.1)	**32.6** (SD 3.2; range: 27.6–40.7)

**Table 2 jcm-10-00057-t002:** Results Study Group, (*n* = 32).

Parameters		Mean	Range
Fluids given	Total praeoperative	440.3 mL	150–700 mL
	0–15 min	319.1 mL	290–450 mL
	16–30 min	305.6 mL	280–350 mL
	31–45 min	319.0 mL	280–400 mL
	46–60 min	331.3 mL	290–400 mL
	Total intraoperative	1275 mL	1150–1500 mL
	Total until end of Surgery	1715 mL	1350–2100 mL
Blood Pressure	MAP ^a^ intraoperative	59 mm Hg	51–76 mm Hg

^a^ MAP (Mean Arterial Pressure).

**Table 3 jcm-10-00057-t003:** Correlations of preoperative hemoglobin levels and hemoglobin levels prior to tourniquet inflation with Hb levels on postoperative day 1 and postoperative day 2.

	Hb POD1	Hb POD2
Preoperative Hb	r_s_ = 0.799, *p* < 0.001	r_s_ = 0.764, *p* < 0.001
Hb prior to surgery	r_s_ = 0.815, *p* < 0.001	r_s_ = 0.658, *p* < 0.001

## Data Availability

The data presented in this study are available on request from the corresponding author.

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
