# Peer review of "The Effect of Hypotensive Anesthesia on Hemoglobin Levels during Total Knee Arthroplasty"

_jcm, 2020, doi:10.3390/jcm10010057_

Round 1

Reviewer 1 Report

Dear authors,
Your manuscript addresses a very interesting topic and presents insights in the intraoperative dynamics of hemoglobin levels during hypotensive epidural anesthesia. The manuscript is clearly structured. The scientific novelty and the potential clinical benefit are obvious.

I only have two minor comments:

1. The authors’ theory of blood dilution is a very plausible explanation for the observations in this study. Since blood volume correlates with the body mass index, did you consider a subgroup analysis of patients with normal vs. high BMI? Patients with normal BMI may be more sensitive to the dilutive effects of lactated Ringer solution.

2. The dilutive effect, of course, also depends on the amount of lactated Ringer solution administered during induction of HEA and tourniquet time. The authors state, that at least 300 ml were given every 15 minutes. However, the manuscript only indicates total fluid amounts during the surgery, and the exact fluid amount administered until the tourniquet is opened is missing. Please specify on this for better interpretation of the obtained results.

Author Response

Thank you very much.

Reviewer 2 Report

Dear Authors, 

The article The Effect of Hypotensive Anesthesia on Hemoglobin
Levels during Total Knee Arthroplasty is really interesting work with important hypothesis in the context of TKA surgery. Nevertheless in the Reviewer opinion it contains some deficiencies, the filling of which could improve the quality of work. The introduction section should be slightly expanded with information about the significance of content from the physiological point of wiev and surgical practice. However, the more important issue concerns the results section that should be present in more detail in the context of examinated patients (gender, age) or measured physiological factors.

Several comments/suggetions:

  • the introduction section is short, it should also contain some more information about physiological significance of hemoglobin level and possible associated problems after TKA surgery 
  • section results is also poor, there is no detailed information about the distribution and impact of gender to the obtained results, correlation of  time elapsed since operation with the level of parameters under consideration; possible correlarion of analysed parameters with age or others physiologocal factors (i.e pressure). Reviewer understand that only the most valuable results are presented, but the others should be at least  commented in short. This is only suggestion, but some addidtional graphic with the presentation  of the most valuable results could improve the quality of this section.

Author Response

Thank you very much.

Round 2

Reviewer 2 Report

Dear Authors,

The Reviewer is satisfied with all the corrections made by the Authors and recommends the article The Effect of Hypotensive Anesthesia on Hemoglobin Levels during Total Knee Arthroplasty for publication in the Journal of Clinical Medicine.